# Coexisting Congenital Mesoblastic Nephroma and Lissencephaly: Unique Case Report with Pathological Analysis and Its Clinical Significance

**DOI:** 10.3390/biomedicines13010196

**Published:** 2025-01-15

**Authors:** Hristina Zakić, Olivera Kontić Vučinić, Jelena Stamenković, Jovan Jevtić, Milena Perišić Mitrović, Maja Životić

**Affiliations:** 1Clinics of Gynecology and Obstetrics, University Clinical Center of Serbia, Faculty of Medicine, University of Belgrade, 11000 Belgrade, Serbia; olja.kontic@gmail.com (O.K.V.); jelenadrstamenkovic@gmail.com (J.S.); milenaperisic@yahoo.com (M.P.M.); 2Institute of Pathology “Dr. Ðorđe Joannović”, Faculty of Medicine, University of Belgrade, 11000 Belgrade, Serbia

**Keywords:** congenital mesoblastic nephroma, lissencephaly, pachygyria, prenatal course

## Abstract

Background: Congenital mesoblastic nephroma represents 3–10% of all pediatric renal tumors. With the advancement of ultrasound diagnostics and magnetic resonance imaging, the diagnosis of this renal neoplasm is increasingly being established prenatally and at birth. It usually presents as a benign tumor, but it can severely affect pregnancy outcomes, contributing to perinatal morbidity and mortality. Lissencephaly belongs to a rare category of neurodevelopmental disorders marked by the absence of a substantial reduction in the typical folds and grooves in the cerebral cortex. The prognosis for patients with lissencephaly is extremely poor, carrying with it a high mortality rate. Case Presentation: We present a case of congenital mesoblastic nephroma (CMN) diagnosed with polyhydramnios at 28 weeks of gestation, which led to preterm delivery at 29 weeks and a fatal outcome for the newborn. Histopathological examination confirmed the diagnosis of CMN along with fetal pachygyria/lissencephaly. The aim of this study is to point out the characteristics and unique correlation between CMN and lissencephaly, and to illustrate the histopathological features of CMN and lissencephaly through an educational example derived from our presented index case. To the best to our knowledge, the association of CMN with lissencephaly has not been described in the literature so far. Conclusions: Outlining the prenatal progression of CMN and the outcome of pregnancies involving fetal CMN and lissencephaly, this case underscores the importance of comprehensive ultrasound examinations, including central nervous system evaluation, to identify potential coexisting anomalies and refine prenatal diagnostic practices.

## 1. Introduction

Renal tumors present at birth are uncommon in general. Among them, the most common are congenital mesoblastic nephroma (CMN), nephroblastoma (Wilms tumor), rhabdoid tumor, and clear cell sarcoma. CMN represents 3–10% of all pediatric renal tumors and is mostly diagnosed within the first three months of life [1,2,3]. However, with advancements in ultrasound diagnostics and magnetic resonance imaging, the diagnosis of this renal neoplasm is increasingly being established prenatally or at birth [1,4].

Ultrasound is the primary tool for the prenatal diagnosis of CMN, and magnetic resonance imaging (MRI) can aid in assessing the origin and morphological characteristics of a fetal abdominal mass [5,6]. Diagnosing CMN before the third trimester of pregnancy is exceptionally challenging, and it is most commonly established in the third or late second trimester. Polyhydramnios emerges as the primary clinical indication in the majority of cases, prompting a comprehensive ultrasound examination during the later part of pregnancy [6,7].

Histopathologically, CMN is classified into three types: classic, cellular, and mixed. It typically exhibits characteristics of a benign tumor and is most effectively managed through surgical removal in neonates [1,2,4]. However, fetal CMN significantly impacts pregnancy outcomes and contributes to perinatal morbidity and mortality, carrying potentially serious complications such as polyhydramnios, premature rupture of membranes, and preterm labor [2,4,7]. Early diagnosis during pregnancy is crucial, as it plays a vital role in shaping the antenatal plan and can influence the management strategy.

Lissencephaly belongs to a rare category of neurodevelopmental disorders marked by the absence or substantial reduction in the typical folds and grooves in the cerebral cortex, leading to an unusually smooth surface [8]. It results from a defective neuronal migration happening at around 12–20 weeks of gestation [9]. The prenatal diagnosis of lissencephaly can be achieved through ultrasound and MRI in the early second trimester [10]. The outlook for patients diagnosed with lissencephaly is quite poor, as a significant proportion do not survive long after birth, or experience postnatal difficulties in daily activities and failure to thrive [9].

We present a case of congenital mesoblastic nephroma (CMN) diagnosed with polyhydramnios at 28 weeks of gestation, which led to preterm delivery at 29 weeks and a fatal outcome for the newborn. Histopathological examination confirmed the diagnosis of CMN along with fetal pachygyria/lissencephaly. To our knowledge, the coexistence of CMN with lissencephaly has not been previously described in the literature. The aim of this study is to point out the distinctive and singular correlation between CMN and lissencephaly, providing a detailed illustration of their histopathological features.

## 2. Case Report

### 2.1. Clinical Presentation

A 22-year-old female patient was admitted as an emergency case to the Clinic of Gynecology and Obstetrics at the University Clinical Center of Serbia in the 28th week of her first pregnancy, which was spontaneously conceived, due to a significantly increased amount of amniotic fluid. Initial ultrasound examination confirmed fetal vitality and appropriate fetal biometrics for gestational age, as well as an increased amount of amniotic fluid (Amniotic Fluid Index: 170 mm). A tumor-like hyperechoic formation in the fetal abdomen, measuring 52.53 × 55.36 mm, was noted but could not be reliably identified (Figure 1).

No fetal breathing or limb movements were detected and end-diastolic block of the umbilical artery was observed. Amnioreduction was performed, evacuating 2000 mL of amniotic fluid. Microbiological and biochemical analysis of the amniotic fluid was normal. During the course of hospitalization, at 29 weeks of gestation, premature rupture of the fetal membranes and spontaneous initiation of labor occurred, resulting in the birth of a live male newborn, weighing 870 g, with an Apgar score of 3 at 5 min. Histopathological examination of the placenta revealed chronic multifocal villitis, acute purulent chorioamnionitis, and subchorionic thrombosis.

After admission to the Neonatal Intensive Care Unit (NICU), the newborn was transferred to the University Children’s Clinic (UCC) for further diagnosis and treatment. The abdomen was markedly protruding above the chest level with a palpable firm swelling in the central and right parts of the abdomen. An ultrasound examination revealed a large intraabdominal tumor mass in the right renal fossa, measuring approximately 52.53 × 55.36 mm. The mass was well delineated from the liver, highly heterogeneous, well vascularized, with a striated zone of calcifications, and it compressed the inferior vena cava (IVC). The renal parenchyma and the right adrenal gland showed no differentiation. Other organs appeared normal. Although neurological findings were consistent with the gestational age, ultrasound examination of the central nervous system (CNS) revealed symmetric cerebral hemispheres with a completely agyric cortex and no dilation of the ventricular system. A subsequent ultrasound examination of the CNS indicated the same gyral pattern with the presence of ventriculomegaly. From the third day of hospitalization, despite the administered therapy, the newborn’s condition worsened, leading to pulmonary hemorrhage. Due to the patient’s severe overall condition, only basic diagnostics of the renal tumor were performed, and surgical treatment was not an option. Despite all implemented measures, the patient remained hemodynamically unstable, with severe metabolic and respiratory acidosis, as well as bradycardia. Cardiopulmonary resuscitation efforts were unsuccessful, and the patient’s death was confirmed on the fifth day of life.

### 2.2. Autopsy Findings

Through clinical autopsy, the pathoanatomical diagnosis was established: the right kidney was almost entirely altered, enlarged, and replaced by tumor tissue that did not breach the renal capsule or infiltrate the adrenal gland. On cross-section, the tissue exhibited a yellowish color interspersed with darker brownish areas (Figure 2).

The tumor of the right kidney was composed of uniform spindle cells arranged in clusters and intermingling fascicles, with indistinct intercellular boundaries. The tumor cells exhibited oval vesicular nuclei and scant eosinophilic cytoplasm, with low, undetectable mitotic activity. Prominent dilated vascular spaces were observed throughout the lesion. The tumor almost completely replaced the normal renal parenchyma. The tumor lacked a capsule, and the tumor–kidney border appeared irregular, with small islands of entrapped non-neoplastic renal tissue (Figure 3). These morphological features were consistent with the classic type of congenital mesoblastic nephroma (CMN).

Immunohistochemically, the tumor cells exhibited diffuse vimentin and INI1 positivity along with focal positivity for WT1, and cyclin D1. The tumor cells were negative for SMA, CD34, bcl-2, ALK, and BCOR. The differential diagnosis considered several spindle cell tumors of the kidney, each with distinct morphological and immunohistochemical features. These morphological and immunohistochemical findings were consistent with a diagnosis of congenital mesoblastic nephroma. However, the differential diagnosis included several spindle cell tumors of the kidney that were carefully evaluated based on their distinct features. Metanephric stromal tumor (MST) was considered due to its spindle cell morphology, but this typically shows alternating nodular cellularity, angiodysplasia, and “onion skinning” around vessels, none of which was present. Additionally, MST is characteristically CD34-positive and may harbor a BRAF V600E mutation, whereas this tumor was CD34-negative. Stromal-type Wilms tumor was another consideration, as it can present as a spindle cell lesion, often containing rhabdomyoblasts or adipose tissue, and is associated with nephrogenic rests. However, these features were not observed in the current tumor, and BCL2, which is often expressed in Wilms tumor, was negative. Moreover, there was no evidence of epithelial or blastemal components upon detailed examination. Clear cell sarcoma of the kidney was excluded due to the absence of its characteristic arborizing vascular pattern and the lack of BCOR expression. Rhabdoid tumor of the kidney, a high-grade spindle cell tumor, was also considered; however, these tumors typically exhibit INI1 loss, whereas the current tumor demonstrated retained INI1 expression. Inflammatory myofibroblastic tumor, another potential differential diagnosis, was ruled out as it is typically ALK-positive, and the current tumor was ALK-negative. The lack of features characteristic of the other entities considered supports the final diagnosis (Figure 3).

The weight of the brain of the prematurely born newborn was 118 g. Furthermore, the gross morphological findings suggested an anomaly in cerebral cortical folding, predominantly of the lissencephalic type (absence of gyri in the large brain), as illustrated in Figure 4, although the possibility of pachygyria (presence of a small number of wide gyri and identified presence of four sulci) could not be completely ruled out with certainty.

Moreover, intraventricular bleeding originating from the periventricular germinal matrix was observed both in the gross analysis, (Figure 4), as well as on the microscopic slides examination. (Figure 5).

Sections taken from the cortex of the large brain did not reveal gyri (the cortex of the large brain was almost completely flat, with only occasional rudimentary sulci). The periventricular germinal matrix was well developed, containing numerous merged fresh hemorrhages. Other organs showed appropriate morphology. The histopathological finding in the lungs indicated diffuse alveolar damage (Figure 5).

### 2.3. Genetic Testing

During hospitalization, an amnioreduction was performed due to a significant increase in amniotic fluid volume, resulting in the evacuation of 2000 mL of fluid. A 10 mL sample was sent for microbiological and biochemical analysis. The QF-PCR analysis of the sample revealed no numerical aberrations in chromosomes 13, 18, 21, X, or Y. The results for informative genetic markers confirmed a diploid number for chromosomes 13, 18, and 21, while the genetic material of the sex chromosome markers indicated a male chromosomal sex. After autopsy, whole exome sequencing of the DNA, performed from paraffin blocks, did not detect any gene mutations potentially causative for the condition, although this cannot be completely ruled out. Three partially relevant mutations were identified, classified as variants of unknown significance. These include heterozygous variants in the autosomal-dominant genes ARID2 and HIVEP2, as well as a hemizygous variant in the OFD1 gene. Although genetic indications suggested targeted testing of the mother for the OFD1 gene variant, given the potential correlation between the genotype and phenotype of the deceased newborn, the parents chose not to undergo genetic testing.

## 3. Discussion

Congenital mesoblastic nephroma was first described by Bolande and colleagues in 1967 as the most common renal neoplasm in newborns and early childhood [11]. Histopathologically, it is classified into three types: classic, cellular, and mixed. The classic form closely resembles leiomyoma, primarily characterized as a solid, firm mass with a yellowish hue, lacking a capsule, featuring indistinct margins and comprising spindle cells in a fusiform arrangement with occasional mitoses. In contrast, the cellular variant, histologically identical to infantile fibrosarcoma, exhibits elevated cell density, increased mitotic activity, necrosis, and hemorrhage. It consists of solid ovoid or fusiform spindle cells with reduced cytoplasm [1,4]. The literary description of the histopathological characteristics of CMN is in correlation with our histopathological findings of the classic type of CMN. The classic type is generally considered a benign tumor. The potential to invade neighboring structures, increased aggressiveness, recurrence, metastasis, and poorer prognosis are associated with the cellular and mixed types, with a survival rate of 85%, compared to 100% for the classic variant [12]. Although we can say that there is no discrepancy between our results and the literature data, given that a live birth occurred in our case, it should be noted that survival in the case of the classic variant of CMN is not always 100%. This pathology carries numerous complications that can lead to a fatal outcome even several days after birth.

Ultrasound is the primary tool for the prenatal diagnosis of CMN. The tumors exhibit solid, unilateral, varied to uniform echogenicity, well-defined boundaries, and significant vascularity [5]. Color Doppler flow can be used to highlight a concentric hyperechoic and hypoechoic ring pattern, known as the vascular “ring” sign, running along the tumor border [6,13,14]. It can displace the abdominal aorta and also move the bowel loops to the right [15]. The cellular type of CMN can be identified on ultrasound and CT scans by its features of cystic areas, hemorrhage, necrosis, and calcification. In contrast, the classic CMN typically presented as a predominantly solid mass [1]. Although the exact origin of the fetal abdominal tumor formation could not be confirmed in our case, the ultrasound image indicated a clearly defined, solid formation with uniform echogenicity, which is consistent with the ultrasound findings of the classic type of CMN as described in the literature. The most prevalent ultrasound finding is polyhydramnios. Specifically, polyhydramnios can be diagnosed several days and even weeks before the detection of the fetal renal mass, making it a potential first clinical manifestation, prompting a comprehensive ultrasound examination during the latter part of pregnancy [6,14,15]. This was also the case in the pregnancy presented here, which was referred to our institution due to increased amniotic fluid levels, initially without a visualized tumor in the fetal abdomen. Polyhydramnios arises from polyuria resulting from elevated renal perfusion and hypercalcemia influenced by the prostaglandins of the kidney tumor. Additionally, it is associated with bowel obstruction induced by the pressure from the renal mass, which almost entirely occupies the abdominal cavity. [6]. Managing polyhydramnios is crucial to prevent premature rupture of the membranes [7]. However, in our case, despite the performed amnioreduction, premature rupture of the amniotic membranes occurred, leading to spontaneous labor, raising the question of whether increased amniotic fluid levels should be managed at all. There have been documented occurrences where CMN appears alongside a typical level of amniotic fluid [1,16]. However, particular attention is warranted in cases of CMN associated with oligohydramnios [17]. CMN may exhibit significant vascularity and be connected to high-output cardiac failure, resulting in hydrops fetalis, a serious indication of a fetal mortality outcome [6].

A characteristic of CMN and simultaneously a challenge for clinicians in prenatal diagnosis is the fact that it is rarely detected during routine second-trimester anomaly screening. Cases have been described where the ultrasound findings were normal from the 20th to the 32nd week of gestation, with CMN being diagnosed later in the course of pregnancy [6,13,14,17]. These findings are consistent with ours, as in our case, there were no indications of any fetal anomalies or abnormalities in the amniotic fluid levels until the 28th week of gestation. The diagnosis is typically established in the third trimester, with an average gestational age of 32 weeks. However, it should be noted that the diagnosis of CMN is most commonly not made prenatally, but rather during infancy, with a median age at diagnosis ranging from 0 days (newborn) to 2.9 months [2]. The primary symptoms observed at the time of diagnosis include the emergence of an abdominal mass, along with hematuria, hypertension, polyuria, and hypercalcemia [1,2,18,19]. We place special emphasis on this moment. Although in our case a formation in the fetal abdomen was prenatally visualized via ultrasound, we could not definitively determine the origin of the tumor formation. Namely, it was only the postnatal ultrasound examination that identified the formation as a tumor originating from the newborn’s kidney. In support of how challenging it can be to establish an accurate diagnosis is the fact that even postnatally, based on ultrasound examination, CMN can be misdiagnosed as a hematoma [20]. In contrast to ultrasound, fetal MRI offers superior soft tissue contrast and various imaging planes, exceling in detecting fetal anomalies, delineating the tumor’s relationship with adjacent structures and identifying the tumor’s origin, size, and characteristics. Consequently, MRI serves as a valuable adjunct to ultrasound in the prenatal diagnosis of CMN and confirmes the ultrasound findings without altering the management plan [6,13,14,15]. Unfortunately, due to the premature rupture of the amniotic membranes, additional diagnostics, such as an MRI of the fetal abdomen, did not take place in our case. Computed Tomography (CT) serves as the primary diagnostic method for identifying kidney tumors and assessing possible metastases. Utilizing CT during pregnancy poses the risk of fetal exposure to radiation, making it an ideal strategy for postnatal tumor evaluation [1,21,22,23,24].

The most commonly reported genetic aberrations in CMN are trisomy 11 and t(12;15)(p13;q25), leading to a fusion of the genes ETV6 and NTRK3, exclusively observed in mixed and cellular type CMN and never in the classic form [2,25]. ETV6 is identified as an oncogene in various leukemias and myeloproliferative syndromes, functioning as a transcription factor. In contrast, NTRK3 belongs to the neurotrophic tyrosine kinase receptor (NTRK) family [26]. The presence of the ETV6-NTRK3 chimeric oncoprotein has demonstrated oncogenicity in various tumor types, including mammary analog secretory carcinoma, secretory breast carcinoma, and papillary thyroid carcinoma [27,28,29]. There is optimism regarding the potential utility of targeted therapy for NTRK-positive congenital mesoblastic nephroma in the neoadjuvant treatment of recurrent or metastatic congenital mesoblastic nephroma [30]. Other recurrently reported genetic aberrations in mixed- and cellular-type CMN include trisomies 8, 17, 20, 7, 18, and 9 [2]. None of the mentioned genetic aberrations was found in our case. Since this is a classic variant of CMN, our results are in correlation with the existing literature. However, whole exome sequencing of the DNA performed from paraffin blocks of the deceased newborn identified three partially relevant mutations: heterozygous variants in the autosomal-dominant genes ARID2 and HIVEP2, as well as a hemizygous variant in the OFD1 gene. AT-Rich Interaction Domain 2 (ARID2) encodes a protein that is part of the SWI/SNF chromatin remodeling complex, which is crucial for regulating gene expression by altering the chromatin structure, thereby enabling or inhibiting transcription. ARID2 is particularly involved in transcriptional regulation linked to cell cycle control, DNA damage response, and tumor suppression. Heterozygous ARID2 mutations have been linked to Coffin–Siris syndrome 6 (CSS6), a rare genetic intellectual disorder characterized by a mild to severe developmental or cognitive delay. ARID2 mutations have been reported in many human cancers including hepatocellular carcinoma, melanoma, urothelial cancer, gastric adenocarcinoma, non-small cell lung cancer, and more [31,32,33]. Human immunodeficiency virus type I enhancer binding protein 2 (HIVEP2) encodes atranscription factor with zinc finger domains that regulates many neurodevelopmental pathways and it has been identified in many neurodevelopmental disorders, intellectual disability, behavioral symptoms, hypotonia, autism spectrum disorders, and microcephaly [34,35,36,37]. OFD1 encodes a protein localized at the centrosome and basal body of primary cilia, playing a crucial role in ciliogenesis, ciliary function, and the establishment of left–right asymmetry. It is encoded by the sequence identified on the X-chromosome. Mutations in OFD1 cause Oral-Facial-Digital Syndrome Type 1, an X-linked dominant male-lethal condition characterized by dysmorphic features affecting the head and face, malformation of the oral cavity, and skeletal, mainly digital, defects [38,39]. Brain developmental anomalies can also be included. Bisschof et al. reported brain MRI findings of the complete OFD1 brain phenotype consisting of complete agenesis of the corpus callosum with large single or multiple interhemispheric cysts, striking cortical infolding of gyri, ventriculomegaly, and moderate to severe cerebellar vermis hypoplasia. Other variable features included hippocampal hypoplasia, reduced volume of white matter, thick anterior commissure, and mild brainstem hypoplasia. Intellectual disability, speech developmental delay, and psychomotor retardation were also reported [39]. Since our case concerns the inadequate development of brain gyri, the association between the mentioned pathogenic gene variants and inadequate brain development in fetuses leaves room for further research in this direction. Future studies should prioritize expanded clinical phenotyping and explore the mechanisms through which pathogenic variants of ARID2, HIVEP2, and particularly OFD1 influence brain development.

Congenital mesoblastic nephroma is considered a tumor with a favorable prognosis by nature. However, serious complications are frequently described with CMN, including polyhydramnios, hydrops fetalis, and spontaneous premature rupture of membranes resulting in preterm birth. These complications are strongly associated with adverse outcomes in newborns, such as respiratory distress syndrome, necrotizing enterocolitis, intraventricular hemorrhage, neonatal hypertension, and the development of metastases [4,6,24]. Drawing a parallel with our case, where despite all applied measures, a fatal outcome occurred due to pulmonary hemorrhage, severe metabolic and respiratory acidosis, and bradycardia, we can conclude that our results align with the literature. This could underscore the importance of preventing preterm birth. Surgery constitutes the primary therapeutic intervention and neonates with live birth demonstrate the capability to lead a healthy life without necessitating adjuvant therapy [1,2,4,6,7]. Given the frequent prenatal complications associated with CMN, we cannot overlook the potential possibility of intrauterine treatment through fetal surgery. Namely, removing or reducing the size of the tumor while the fetus is still in utero, could possibly prevent or reduce postnatal complications and improve the neonatal outcome. Considering that CMN is typically diagnosed later in pregnancy, when it has already reached a substantial size, potential fetal surgery could face significant challenges. The tumor might be challenging to operate on or unreachable due to its location within the fetal kidney or the surrounding tissues, carrying complications such as fetal injury, preterm labor, or fetal loss. Further research, improvements in surgical techniques, and a deeper understanding of long-term outcomes are essential before this can become a standard approach for treating this rare condition. Moreover, the risks and benefits should be carefully evaluated on an individual basis. The efficacy and safety of fetal surgery for CMN remain uncertain due to the absence of large-scale studies and long-term follow-up data.

When it comes to lissencephaly, the term is derived from the Greek words “lissos”, meaning smooth, and “enkephalos”, meaning brain. It comes with an estimated prevalence of 1 in 100,000 births. Embryologically, it results from a defective neuronal migration, along with more complex and subtle anomalies affecting cell proliferation and differentiation, neurite outgrowth, axonal pathfinding transport, connectivity, and myelination [8,9]. It has been divided into two categories: classic lissencephaly (type 1 lissencephaly or agyria-pachygyria) and cobblestone complex (type 2 lissencephaly). So far, more than 19 genes associated with lissencephaly have been identified. The most common of them are LIS1, DCX, TUBA1A, RELN, VLDLR, ARX, WDR62, and the PAFAH1B1 gene [8,9]. The functions of certain lissencephaly genes, including LIS1, DCX, and TUBA1A, are closely associated with microtubules, which, as integral components of the cytoskeleton, play crucial roles in cellular processes such as mitosis and cytokinesis. The prenatal diagnosis of lissencephaly can be achieved through ultrasound and magnetic resonance imaging in the early second trimester, revealing the abnormal development of sulci and gyri [10]. Prenatal ultrasonography features commonly involve commissural anomalies, ventriculomegaly, hydrocephalus, asymmetric ventricles, cerebellar vermian or hemispheric anomalies, and abnormal head circumference [10]. Although the literature supports the possibility of a prenatal ultrasound diagnosis of lissencephaly, that was not the case in our situation. Since the course of the pregnancy required a focus on polyhydramnios and the prevention of premature rupture of the membranes, detailed ultrasound evaluation of the fetal CNS was not performed. The complete fetal biometry was adequate for the given gestational age; however, the postnatal results revealed a discrepancy. MRI can detect fetal lissencephaly between 20 and 24 weeks of gestation. However, false positives are possible, especially during the second trimester [40]. The prognosis for individuals with lissencephaly is notably grim, as a significant proportion do not survive long after birth, or experience postnatal failure to thrive. The weight of the brain of the prematurely born newborn in our case was 118 g. Compared to standard fetal autopsy values, the brain weight corresponded to the period of 25 to 27 weeks of gestation. The brain of the newborn in our case had a lower weight than expected for that gestational age [41]. Considering the availability of literature data stating that at least one sulcus appears per week after the 20th week of gestation, and given that we identified the presence of four sulci, the fetal brain in our case would correspond to 24 weeks of gestation [41,42]. Upon searching the literature, we did not encounter a case similar to ours, neither when considering a potential genetic factor, nor when considering an association with a similar condition. However, in a 2019 study, a case of CMN was described where signs of microcephaly, hypotonia, and developmental delay were noted at 6 months of age. An MRI of the brain performed at this age revealed cortical and subcortical atrophy, indicative of prolonged hypoxia. By the age of 1, the clinical condition was characterized by severe microcephaly, right-sided tetraparesis, global hypotonia, and developmental delay [4].

## 4. Conclusions

Congenital mesoblastic nephroma (CMN) often manifests in late pregnancy, characterized by rapid tumor growth, as observed in our case. The substantial tumor size at diagnosis may result from its echogenic similarity to normal renal pyramids, complicating differentiation during prenatal ultrasound. The tumor’s proximity to normal parenchyma, absence of a distinct capsule in smaller masses, and blending with surrounding kidney tissue further obscure detection. Early-stage tumors are often undetectable due to their small size, but abnormal renal enlargement or asymmetry warrants careful examination of the kidneys, adrenal glands, and contralateral kidney. While detailed ultrasound remains the gold standard for CMN diagnosis, MRI can provide additional clarity in cases of diagnostic uncertainty. The prenatal detection of CMN is crucial for optimal management, improved outcomes, and informed parental counseling. Emphasis should be placed on thorough evaluation of the fetal urinary tract, especially in polyhydramnios cases, requiring close monitoring to prevent preterm birth. Although genetic testing in our case did not confirm a specific genetic condition, targeted testing of the OFD1 gene in the mother is warranted due to potential genotype–phenotype correlations with the deceased newborn. Our investigation did not identify a genetic link between CMN and lissencephaly, a previously unreported coexistence. This case underscores the importance of comprehensive ultrasound examinations, including central nervous system evaluation, to identify potential coexisting anomalies and refine prenatal diagnostic practices.

## Figures and Tables

**Figure 1 biomedicines-13-00196-f001:**
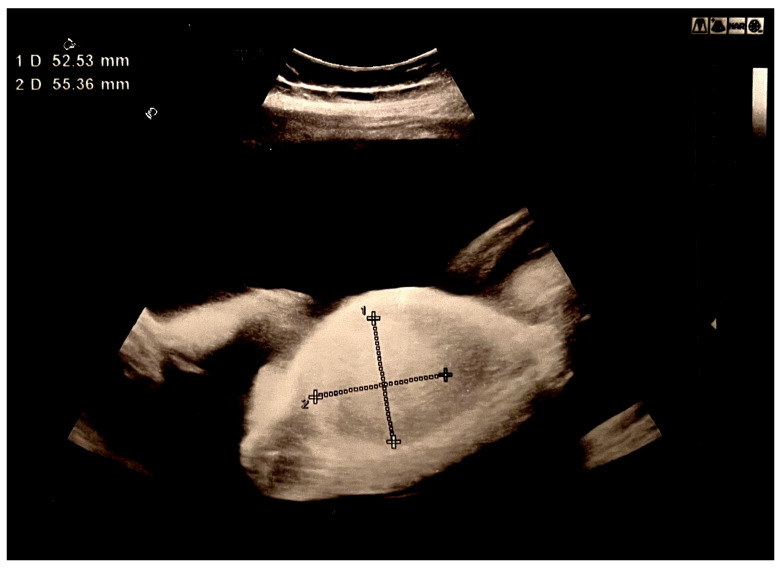
Hyperechoic tumor formation in the abdomen of the fetus, size 50 × 50 mm.

**Figure 2 biomedicines-13-00196-f002:**
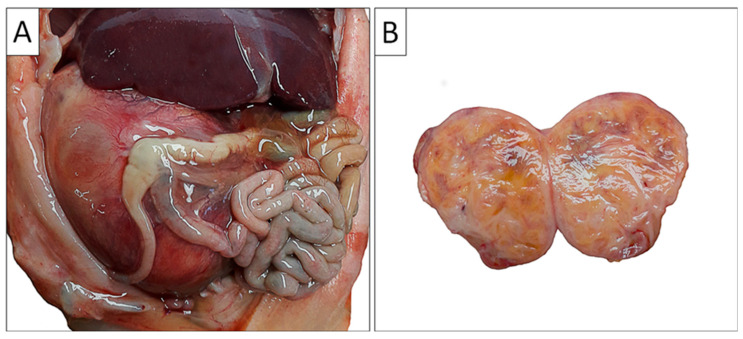
**Gross autopsy findings of the abdominal cavity and the right kidney**. (**A**) Enlarged right kidney compresses abdominal organs (liver and intestinal tract); (**B**) Cross-section of the right kidney illustrates tumor mass which completely replaced the normal kidney parenchyma.

**Figure 3 biomedicines-13-00196-f003:**
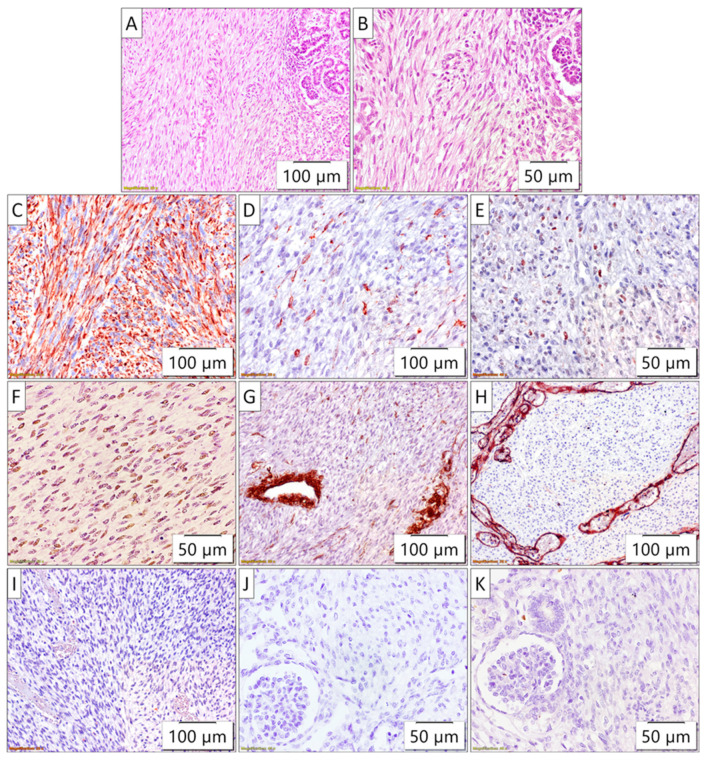
**Pathohistology of congenital mesoblastic nephroma**. (**A**,**B**) The tumor was composed of uniform spindle cells arranged in clusters and fascicles, with indistinct intercellular boundaries, trapping small islands of non-neoplastic renal tissue (HE stain); immunohistochemically, the tumor cells exhibited diffuse (**C**) vimentin and (**F**) INI1 positivity along with focal positivity for (**D**) WT1, and (**E**) cyclin D1. The tumor cells were negative for (**G**) SMA (internal positive control are blood vessels), (**H**) CD34 (internal positive control are blood vessels), (**I**) bcl-2, (**J**) ALK, and (**K**) BCOR.

**Figure 4 biomedicines-13-00196-f004:**
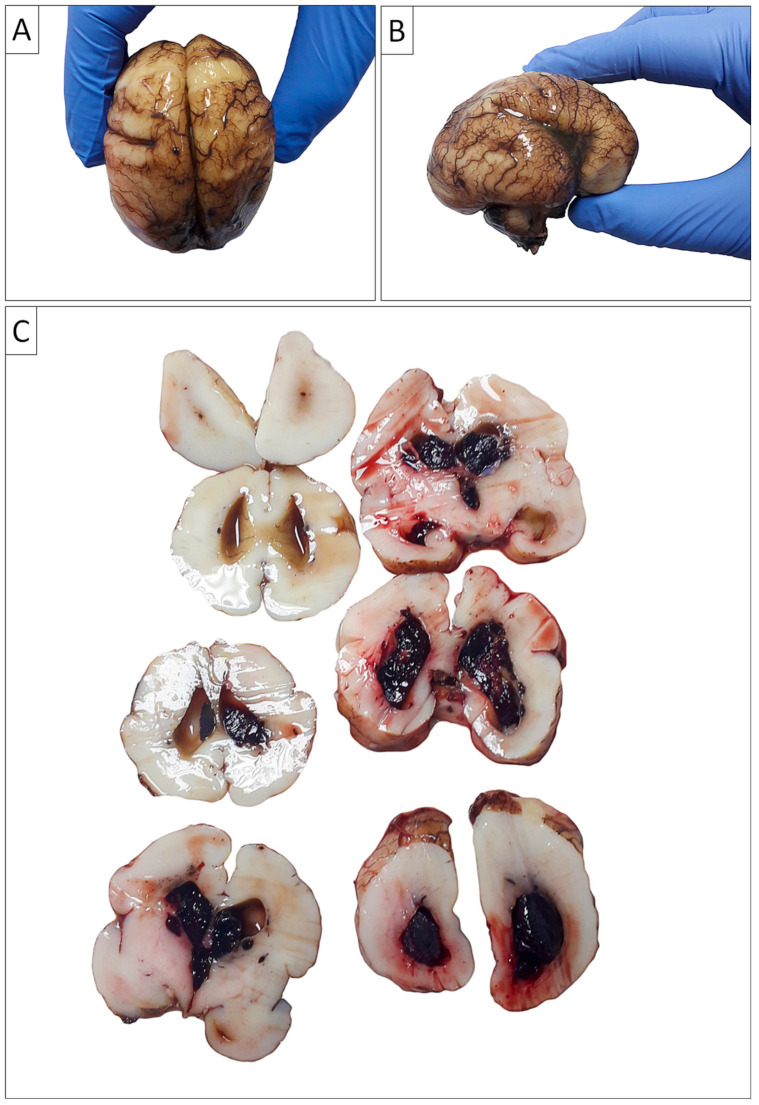
**Gross autopsy findings of the brain**. (**A**) Antero-posterior view of left and right cerebral hemispheres revealed abnormal cortical folding, predominantly of the lissencephalic type; (**B**) Temporal-parietal view of the right cerebral hemisphere revealed abnormal cortical folding, predominantly of the lissencephalic type, although the possibility of pachygyria cannot be completely ruled out with certainty; (**C**) Coronal planes of brain sections revealed massive intraventricular bleeding.

**Figure 5 biomedicines-13-00196-f005:**
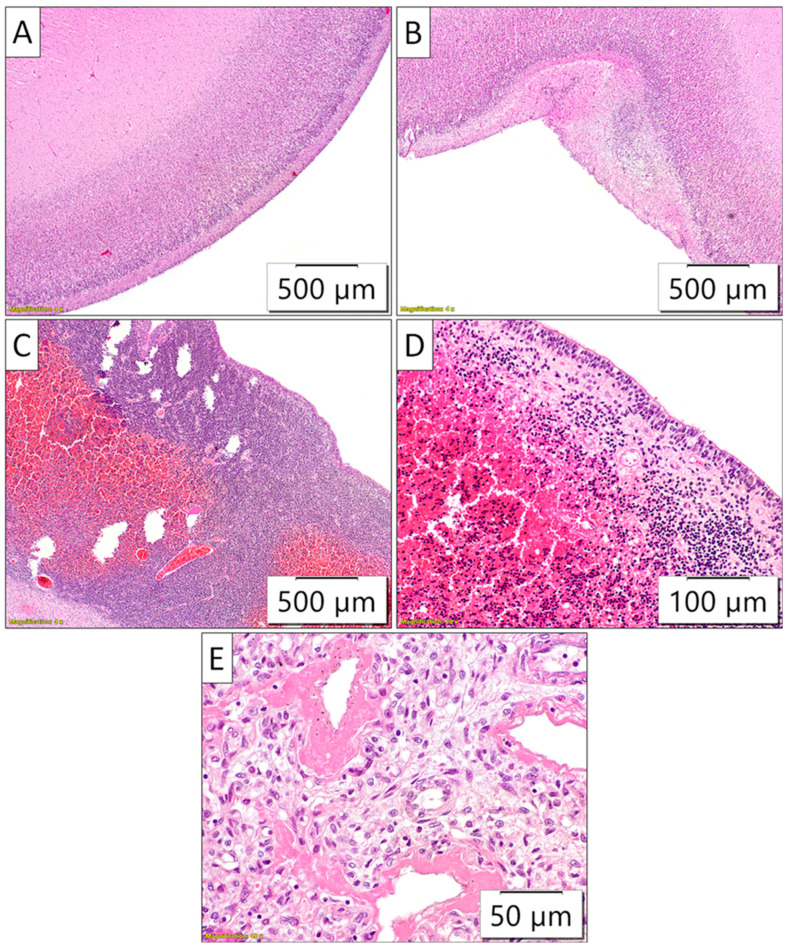
**Pathohistology of the brain and lung sections.** (**A**) The absence of large brain gyri; thus, the brain surface was almost completely flat; (**B**) Occasional rudimentary sulci of the large brain cortex. (**C**,**D**) Well-developed periventricular germinal matrix with pronounced fresh hemorrhages; (**E**) Diffuse and thick intraalveolar hyaline membranes illustrated the morphology of severe diffuse alveolar damage.

## Data Availability

The original contributions presented in this report are included in the article. Further enquiries can be directed to the corresponding authors.

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
