# Peer review of "Coexisting Congenital Mesoblastic Nephroma and Lissencephaly: Unique Case Report with Pathological Analysis and Its Clinical Significance"

_biomedicines, 2025, doi:10.3390/biomedicines13010196_

Round 1

Reviewer 1 Report

Comments and Suggestions for Authors

The manuscript entitled "Association of congenital mesoblastic nephroma and cerebral lissencephaly/pachygyria: case report and literature review" is well written and covers the theme of nephrotic tumors and other abnormalities during fetal development. In general, the manuscript is well written in clear understandable English. Nevertheless, I have some commentaries and suggestions.

Major

1. I recommend for the authors to change the title. By the reading of the current title it may get the impression that there is correlation between nephroblastoma and lissencephaly, however it is not truth.

2. Which was the weight of the brain of the prematurely newborn child? Was it normal?

3. You mention the mutation in genes ARIS2, HIVEP2 and OFD1. Please, indicate which are the functions of that genes.

4. In the figures were you provide the brain of newborn, please, add the figures of the normal brain of the fetuses at the same age. Maybe there are some figures from the atlases. It will be interesting to see that comparison.

5. Discuss the possibility and limitations of intrauterine tretment of fetal neuroblastoma by fetal surgery, for example.

Minor

1. Indicated in the Introduction section the frequency of fetal nephroblastoma in population. It is very rare disease.

2. Line 285 typeset.

Reviewer 2 Report

Comments and Suggestions for Authors

Congenital mesoblastic nephroma (CMN) accounts for 3-10% of pediatric renal tumors and is increasingly diagnosed prenatally due to advancements in ultrasound and MRI, though it can lead to serious pregnancy complications. This study highlights the rare association between CMN and lissencephaly, emphasizing the importance of comprehensive prenatal ultrasound, including central nervous system evaluation, to identify coexisting anomalies and improve diagnostic practices.

Major comments

1. The authors should add a clear description of what is the same and what is different in this case from previous cases.

2. If you reviewed articles, you should describe how you extracted references from how many articles and by what method, otherwise it is not a literature review.

3. If the disease is caused by genetic changes, the interpretation of genetic test results should be cautious, and the VUS results should be presented as a waveform using the Sanger method or other methods as validation of the exome test results.

Minor comments

1. The histopathology of the CMN in this case should specify whether it is classic, cellular, or mixed type.

2. Abstract: line 22; “decribed” should be “described”.

3. Figure 1 has poor resolution and should be replaced with one of better resolution.

Reviewer 3 Report

Comments and Suggestions for Authors

Brief summary:

The aim of the paper “Association of congenital mesoblastic nephroma and cerebral lissencephaly/pachygyria: case report and literature review” is the presentation of a rare case, added with ultrasound, histopathological, immunohistochemistry, and genetic testing.

Broad comments:

The manuscript is well organized into 4 sections, being based on 31 articles. The English language is fairly good, data are illustrated in five figures of good quality.

Specific comments:

The sections and subsections of the manuscript are appropriate.

It is recommended to provide more information regarding the complying to the ethics principles.

Minor English revision should be performed, such as “examionations” (in the Abstract section) and the use of present tense mixed with past tense (in the Autopsy findings).

In the last section, of Discussions, the author highlights the value of this case presentation, in the context of other reports. It is recommended to compare the characteristics of the reported congenital mesoblastic nephroma with those of other cases reported in the literature.

Round 2

Reviewer 2 Report

Comments and Suggestions for Authors

The authors replied to my questions.